# Prevalence of Different Courses of Feline Leukaemia Virus Infection in Four European Countries

**DOI:** 10.3390/v15081718

**Published:** 2023-08-10

**Authors:** Juliana Giselbrecht, Stéphanie Jähne, Michèle Bergmann, Marina L. Meli, Benita Pineroli, Eva Boenzli, Svenja Teichmann-Knorrn, Yury Zablotski, Maria-Grazia Pennisi, Nicolas Layachi, Rodrigo Serra, Stefano Bo, Regina Hofmann-Lehmann, Katrin Hartmann

**Affiliations:** 1LMU Small Animal Clinic, Centre for Clinical Veterinary Medicine, 80539 Munich, Germany; steffijaehne@gmail.com (S.J.); n.bergmann@medizinische-kleintierklinik.de (M.B.); y.zablotski@med.vetmed.uni-muenchen.de (Y.Z.); hartmann@lmu.de (K.H.); 2Clinical Laboratory, Department of Clinical Diagnostics and Services, and Centre for Clinical Studies, Vetsuisse Faculty, University of Zurich, 8057 Zurich, Switzerland; mmeli@vetclinics.uzh.ch (M.L.M.); bpineroli@vetclinics.uzh.ch (B.P.); eva.boenzli@uzh.ch (E.B.); rhofmann@vetclinics.uzh.ch (R.H.-L.); 3Veterinary Clinic Oberhaching, 82041 Oberhaching, Germany; teichmann-knorrn@tierklinik-oberhaching.de; 4Department of Veterinary Sciences, University of Messina, 98168 Messina, Italy; mariagrazia.pennisi@unime.it; 5Layachi Veterinary Clinic, 33300 Bordeaux, France; layachivet@yahoo.fr; 6Investigacao Veterinaria Independente, 1700-119 Lisbon, Portugal; rodserra@gmail.com; 7Ambulatorio Veterinario Bo-Ferro, 10123 Turin, Italy; stefano@veterinariassociati.it

**Keywords:** FeLV, retrovirus, prevalence, p27 antigen, proviral DNA, viral RNA, antibody levels, Europe

## Abstract

Prevalence of progressive feline leukaemia virus (FeLV) infection is known to still be high in cats in Europe, especially in Southern Europe, but the prevalence of other outcomes of FeLV infection has not been determined in most countries. The present study aimed to investigate the prevalence of progressive, regressive, abortive, and focal infection in four European countries, two with a high (Italy, Portugal) and two with a low expected prevalence (Germany, France). Blood samples of 934 cats (Italy: 269; Portugal: 240; France: 107; Germany: 318) were evaluated for the p27 antigen, as well as anti-whole virus, anti-SU, and anti-p15E antibodies by enzyme-linked immunosorbent assay (ELISA) in serum and for proviral DNA by quantitative polymerase chain reaction (qPCR) in whole blood. Positive p27 antigen ELISA results were confirmed by reverse transcriptase-qPCR (RT-qPCR) detecting viral RNA in saliva swabs and/or blood. The outcome of FeLV infection was categorised as progressive (antigen-positive, provirus-positive), regressive (antigen-negative, provirus-positive), abortive (antigen- and provirus-negative, antibody-positive), and focal (antigen-positive, provirus-negative) infection. Overall FeLV prevalence was 21.2% in Italy, 20.4% in Portugal, 9.5% in Germany, and 9.3% in France. Prevalence of progressive, regressive, abortive, and focal infection in Italy was 7.8%, 4.5%, 6.3%, and 2.6%; in Portugal 3.8%, 8.3%, 6.7%, and 1.7%; in Germany 1.9%, 1.3%, 3.5%, and 2.8%; in France 1.9%, 3.7%, 2.8%, and 0.9%, respectively. In conclusion, overall FeLV prevalence is still very high, especially in Southern European countries. Therefore, testing, separation of infected cats, and vaccination are still important measures to reduce the risk of FeLV infection.

## 1. Introduction

Feline leukaemia virus (FeLV) is a gammaretrovirus that is widespread worldwide and one of the most important infectious agents in cats [1,2,3]. Due to the complex pathogenesis and the different courses of FeLV infection, diagnosis is challenging and often not possible using a single test. FeLV infection can take progressive, regressive, abortive, or focal (atypical) courses [1,2]. However, even when established, courses can change into each other. For example, cats that are initially progressively infected can develop a regressive course of infection. Conversely, regressively infected cats can become progressively infected. Differentiation between the FeLV outcomes is difficult, especially in naturally infected cats [1,2,3,4]. The individual outcome in a FeLV-infected cat is determined by the immune status of the infected cat, influenced by pre-existing immunity or age, and by viral characteristics, such as the virulence of the virus or infection pressure. Several factors, such as immunosuppression, coinfections, and stress can influence the immune response, and thus the course of infection [2]. In progressive infection, the immune system of the affected cats is unable to sufficiently control virus replication and its systemic spread, and viraemia persists. During the viraemic phases, free p27 antigen can be detected in serum/plasma, proviral DNA (deoxyribonucleic acid) in blood, and viral RNA (ribonucleic acid) in blood and saliva [5]. Progressive infection can lead to immunodeficiency, bone marrow suppression, and neoplasia, and is commonly fatal [4,6,7]. On the contrary, with the help of an effective immune response, cats that are regressively infected are able to stop or significantly inhibit viral replication. Due to the pronounced immune response, regressively infected cats generally have high levels of virus-neutralising antibodies. In contrast to progressively infected cats, in regressively infected cats, viraemia never occurs or only lasts briefly at the beginning of the infection and potentially (rarely) reoccurs later, after reactivation [6]. Abortively infected cats produce virus-neutralising antibodies and are able to effectively control virus replication [8,9,10]. Neither FeLV p27 antigen, proviral DNA, nor viral RNA can be detected in these cats. Abortive infection can only be diagnosed by the detection of antibodies [4,9,11,12,13]. 

FeLV prevalence of progressive FeLV infection, which is easily detected, varies worldwide, ranging from 1 to 9% in Europe [14]. According to a recent Europe-wide study of the Advisory Board on Cat Diseases [15] including 6005 cats in 30 European countries, the highest prevalence was found in Portugal (8.8%), Hungary (5.9%), Italy (5.7%), and Malta (5.7%). France and Germany were considered to be low-prevalence countries, with a prevalence of 1.0% and 0.3%, respectively [14]. In this and many other prevalence studies, however, only progressive infections were assessed. Nevertheless, when considering all courses of FeLV infection, the overall FeLV prevalence is considered to be much higher. This was demonstrated in a German study in 2012, in which 1.8% (9/495) of cats were progressively, 1.2% (6/495) regressively, and 9.2% (22/246) abortively infected with FeLV [12]. However, the prevalence of regressive and abortive infection is largely unknown in most European countries.

Therefore, the aim of the present multicentre, prospective, and cross-sectional study was to determine the prevalence of all courses of FeLV infection in cats from four different countries in Europe with different FeLV prevalence, including two countries with high suspected prevalence (Italy and Portugal) and two countries with low suspected prevalence (Germany and France).

## 2. Materials and Methods

### 2.1. Samples and Data Collection

A total of 934 cats from four different European countries were prospectively included. Two countries with a suspected high overall FeLV prevalence (Italy and Portugal) and two countries with a suspected low overall FeLV prevalence (Germany and France) according to a previous study [14] were included. Of all cats, 318 originated from Germany, 269 from Italy, 240 from Portugal, and 107 from France. A body weight of >1 kg was a prerequisite for inclusion. Overall, eleven veterinary clinics in Italy, four veterinary clinics in Portugal, two veterinary clinics in Germany, and one veterinary clinic in France participated. Cats from which blood was collected for different reasons were included and originated from private homes, shelters, or were stray. The study was accepted by the ethical committee of the Centre for Clinical Veterinary Medicine of the LMU Munich, Germany (reference number 142–25–08–2018).

Serum and ethylenediaminetetraacetic acid (EDTA) whole blood samples were collected from all cats. In addition, saliva swabs (cotton swabs with plastic shafts) were collected from 815/934 cats and transferred to a reagent tube (1.5 mL, Sarstedt, Nümbrecht, Germany). Both blood and saliva samples from Italy, Portugal, and France were stored at −20 °C, while samples from Germany were stored at −80 °C, in all cases for a maximum of 24 months before the samples were sent on dry ice to the Clinical Laboratory from the Department of Clinical Diagnostics and Services, and the Center for Clinical Studies at the Vetsuisse Faculty, University of Zurich, for examination. In Zurich, all samples were stored at −80 °C until analysis for a maximum of four months.

Data were provided from 777/934 owners using a questionnaire that included questions on cat identification (name of the veterinary practice, cat and cat owner, date of collection) and demographic data of the cat (age, sex, neutering status, cat breed), on housing conditions (indoor only, outdoor only, or indoor and outdoor access, number of cats in the household), history of illness, previous FeLV infection status if available (FeLV antigen-positive, FeLV antigen-negative, unknown), and FeLV vaccination status.

### 2.2. Sample Analysis

#### 2.2.1. Detection of FeLV p27 Antigen

All 934 cats were tested for the presence of free p27 antigen in serum by the sandwich enzyme-linked immunosorbent assay, as described previously [16]. All samples were tested in duplicates and absorbances were read using a microplate reader (Synergy H1, Biotek, Winooski, VT, USA). Values > 4% of the positive control were considered positive [17].

#### 2.2.2. FeLV Viral RNA in Saliva and Blood

To confirm positive p27 antigen results, blood and saliva samples from all p27 antigen-positive cats were tested for FeLV viral RNA using a published reverse transcription quantitative polymerase chain reaction (RT-qPCR) assay [14,18]. Positive and negative controls were run in parallel with each RT-qPCR. All negative samples were retested, and diluted at 1:5 and 1:10 in a neutral buffer with a pH of 7.4 (0.15 M sodium chloride, 1 mM EDTA, 0.05 M Tris-base, 0.1% BSA, 0.1% Tween 20) to make possible inhibition unlikely. The viral RNA copy number was determined using a standard curve (serial tenfold dilutions of purified RNA from FEA/FeLV-A infected cells [18]) run with the same conditions (reaction composition, instrument and thermal profile) as described [14].

#### 2.2.3. FeLV Proviral Load in Blood

FeLV proviral DNA was evaluated in all 934 cats. Total nucleic acids were extracted from 100 µL EDTA anticoagulated whole blood using the MagNA Pure 96 instrument (Roche Diagnostics AG, Rotkreuz, Switzerland) and Viral NA SV Kit (Roche Diagnostics AG, Rotkreuz, Switzerland) according to the manufacturer’s instructions with 100 µL elution buffer [10]. For all samples, the viral nucleic acid (NA) plasma external lysis SV 4.0 protocol was applied and negative controls of phosphate-buffered saline (PBS) run in parallel with each batch of samples to monitor for cross-contamination. Proviral DNA was amplified and quantified using 5 μL of TNA and 20 µL of DNA quantitative PCR (qPCR) Mastermix (Eurogentec, Seraing, Belgium) containing 480 nM primers (exoFeLV-U3F2, exoFeLV-U3R3) and a 160 nM probe (exoFeLV-U3p). All oligonucleotides were synthesised by a Microsynth AG (Balgach, Switzerland). The temperature profile consisted of 2 min (min) at 50 °C, denaturation of 10 min at 95 °C, followed by 45 cycles of 95 °C for 15 s (s) and 60 °C for 1 min. The FeLV proviral copy numbers in the single samples were determined by coamplifying 10-fold serial dilutions of a DNA standard template, as described [18]. All samples that tested positive in the p27 antigen ELISA were diluted 1:5 and 1:10 and retested to avoid a false-negative result in the provirus qPCR due to possible inhibition. To verify the quantity and quality of the TNA, a qPCR for feline albumin was performed on all 934 TNA samples [19].

#### 2.2.4. Detection of Anti-FeLV Antibodies

Serum samples of all 934 cats were analysed for the presence of antibodies against FeLV whole virus (FL-74), FeLV SU (p45), and FeLV p15E using indirect ELISA, as previously described [4,10,20].

##### Detection of Anti-Whole Virus (FL-74) and Anti-SU (p45) Antibodies

Serum samples were analysed for the presence of antibodies to FeLV p45 and to FeLV whole virus by ELISA, using 100 ng of the p45 antigen and 100 ng of gradient-purified FL-74 FeLV per well and serum dilutions of 1:200, respectively, as described previously [21]. Positive and negative controls were run with each assay. For negative controls, serum samples from specific pathogen-free (SPF) cats were used. For positive controls, serum samples collected from cats known to have antibodies against FeLV (=100% positive control) were used. Values ≥25% of the positive control were considered positive [12].

##### Detection of Anti-p15E Antibodies

An ELISA for the detection of anti-p15E antibodies was used, as previously described [10]. Positive and negative controls were included in each assay. For negative controls, serum from SPF cats was used. For positive controls, pooled serum samples from cats experimentally infected with FeLV-A/Glasgow-1 (=100% positive control) were included. Relative optical density (ROD) values were determined using the formula ROD = [(sample OD − negative control OD)/(positive control OD − negative control OD)]. Samples with an ROD that tested >16.3% (ROD value 0.163) compared to the positive control were considered anti-p15E antibody-positive [10].

### 2.3. FeLV Infection Status

According to the European Advisory Board on Cat Diseases [15], the different courses of FeLV infection were defined as follows [3]: (1) progressive infection (p27 antigen-positive, provirus-positive), (2) regressive infection (p27 antigen-negative, provirus-positive), (3) abortive infection (p27 antigen-negative, provirus-negative, anti-p15E- and anti-SU antibody-positive), (4) focal infection (p27 antigen weakly positive, provirus-negative), and (5) FeLV-uninfected (p27 antigen-negative, provirus-negative, anti-p15E and anti-SU antibody-negative) (Table 1). Samples that were p27 antigen-negative, provirus-negative, anti-p15E antibody-positive, but anti-SU antibody-negative were considered “unclassified”.

### 2.4. Statistical Analysis

Laboratory data of all cats were analysed using Excel (Microsoft Germany GmbH, Munich, Germany) and the R statistical language (version 4.1.2; R Core Team, 2020). The normality of data distribution was determined by the Shapiro–Wilk test. Due to non-normally distributed data, a nonparametric test was used for statistical analysis. The Mann–Whitney U test was used to compare p27 antigen concentration in progressively and focally infected cats. In addition, Mann–Whitney U test was used to compare the proviral load of progressively and regressively infected cats. The Kruskal–Wallis and Dunn–Bonferroni tests were used to compare anti-FeLV antibody levels in progressively, regressively, abortively, and focally infected cats. The Spearman correlation was conducted for the correlation between p27 antigen concentration in blood, viral RNA loads in blood and saliva, and anti-p15E antibody levels. Results with a *p*-value <0.05 were considered statistically significant.

## 3. Results

### 3.1. Study Population

At the time of sampling, 56% (377/670) of cats were healthy and had no history of illness, whereas 44% (293/670) of cats were either sick at the time of testing or had a history of illness. According to the questionnaires, in 70% (654/934) of all cats, FeLV vaccination status was known; 12% (81/654) of these cats were vaccinated against FeLV. The vaccine manufacturers and vaccination dates were available in 67% (54/81) of the vaccinated cats. Age was known in 82% (765/934) of the cats and ranged from 8 weeks to 20 years (median: 3 years). Table 2 summarises the signalment, history of illness, vaccination, and housing condition of the cats. Table 3 gives an overview of the vaccines used in the individual countries.

### 3.2. FeLV Test Results

A summary of testing results in the different countries is shown in Table 4.

### 3.3. Prevalence of Different FeLV Courses in the Four European Countries

Overall, considering all courses, FeLV infection prevalence was 15.6% (146/934), with 20.8% (56/269) in Italy, 20.4% (49/240) in Portugal, 9.3% (10/107) in France, and 9.1% (29/318) in Germany (Table 5). Not all cats could be assigned to a definitive course of infection; cats with an unknown vaccination status that were p27 antigen-negative, provirus-negative, anti-p15E antibody-positive but anti-SU antibody-negative were considered “unclassifiable” (total 2.1% (20/934); Italy, 2.2% (6/269), Portugal 0.8% (2/240), Germany 1.3% (4/318), and France 7.5% (8/107)) and were not included in the total infection prevalence.

### 3.4. Viral Loads and p27 Antigen Levels in the Different Courses of Infection

All cats that tested positive for p27 antigen by ELISA were considered to have either a progressive or a focal infection. Cats with progressive infection had significantly higher antigen levels (median: 91%, range: 5–197%) compared to cats with focal infection (median: 6%, range: 4–20%, *p* < 0.001) (Figure 1).

The blood proviral loads (Figure 2) were significantly higher in progressively infected cats (n *=* 37) (range: 43–9,501,447 copies) than in regressively infected cats (n *=* 40) (range: 1–428 copies). Two cats with a progressive course had a low proviral load (copy numbers < 60 (43, 59)). All other progressively infected cats (n *=* 35) had high proviral loads (copy numbers > 60 (237–9,501,447 copies). Among the regressively infected cats, one cat had a high proviral load (428 copies). All other regressively infected cats (n *=* 39) had low proviral loads (copy numbers < 60 (1–59 copies)).

In cats with progressive infection, a strong significant correlation was observed between the p27 antigen concentration and viral RNA load in saliva (Spearman correlation = 0.65; *p* < 0.001) (Figure 3A). There was also a correlation between the p27 antigen concentration and viral RNA load in blood (Spearman correlation = 0.38; *p* = 0.027) (Figure 3B).

### 3.5. Antibody Levels in Different Courses of Infection

Cats with an abortive infection had significantly higher anti-SU (p45) antibody levels (median: 37%, range: 25–116%) compared to cats with progressive (median: 15%, range: 0–62%) (*p* < 0.001), regressive (median: 20%, range: 1–113%) (*p* < 0.001), and focal (median: 23%, range: 6–53%) (*p* < 0.001) infection. Regressively infected cats had significantly higher anti-whole virus antibody levels (median: 49%, range: 0–190%) compared to progressively (median: 4%, range: 0–63%) (*p* < 0.001) and abortively (median: 18%, range: 0–82%) (*p* = 0.011) infected cats. Progressively infected cats had significantly lower antibody levels compared to abortively infected (*p* = 0.005) and focally infected cats (median: 23%, range: 5–123%) (*p* ≤ 0.001). In contrast, progressively infected cats had significantly higher anti-p15E antibody levels (median: 64%, range: 0–121%) than abortively (median: 25%, range: 17–114%) (*p* < 0.001) infected cats. Abortively infected cats had significantly higher antibody levels compared to regressively (median: 19%, range: 0–152%) (*p* < 0.001) and focally infected cats (median: 11%, range: 0–40%) (*p* = 0.002) (Figure 4). In addition, there was a significant correlation between anti-p15E antibody levels in blood and p27 antigen in the blood (Spearman correlation = 0.4709; *p* ≤ 0.001) (Figure 5).

## 4. Discussion

This is the first study to determine the FeLV prevalence in four European countries considering all courses of FeLV infection (progressive, regressive, abortive, focal (atypical)). There was still a larger number of infected cats than expected when considering all infection stages, with more cats being infected in Southern Europe (Italy 21.2%, Portugal 20.4%) than in Western Europe (Germany 9.5%, France 9.3%).

In the present study, more cats in Southern Europe (Italy 7.8%, Portugal 3.8%) had a progressive course of infection in contrast to cats in Western Europe (Germany 1.9%, France 1.9%). Similar results on FeLV prevalence have been obtained in a pan-European study conducted by the Advisory Board of Cat Diseases [15] in 2019, in which 6005 cats were tested for FeLV shedding by RT-qPCR from saliva swabs [14]. The prevalence in Portugal, Italy, Germany, and France was 8.8%, 5.7%, 0.3%, and 1.0%, respectively. One possible explanation why the prevalence of progressively infected cats in Germany and France was higher in the present study compared to the study in 2019 could be that, in Germany in particular, several cats from animal shelters were sampled, and FeLV prevalence could be higher in shelter cats compared to those presented at veterinary clinics [22]. In France, only one veterinary clinic, and therefore only a small proportion of the country, was included in the study. The prevalence might be lower in other parts of France. It is unlikely that the higher prevalence in the present study is caused by false-positive results as in the present study, all progressively infected cats also tested positive for viral RNA in blood; and all progressively infected cats in which a saliva swab was available were positive for viral RNA in RT-qPCR. In a study in Switzerland including blood samples from 445 cats, the sensitivity and specificity for the detection of FeLV RNA in saliva by RT-qPCR were 98.1% and 99.2%, respectively, and the detection of viral RNA in saliva and viral RNA in blood, each tested by RT-qPCR, showed almost perfect agreement (kappa = 0.96) [5].

In focally (atypically) infected cats, there are usually little replicating virus and infected cells in the peripheral blood. However, focally infected cells can produce soluble p27 antigen and release it into the peripheral blood, which can lead to a positive p27 antigen test [11,23,24,25]. In the present study, viral RNA was also detected in saliva in these cats. Therefore, detection of RNA in the saliva of focally infected cats is possible when the replication of the virus after oronasal ingestion is limited to the mucosa of the oropharynx and local lymphoid tissue and affects the epithelium of the salivary glands and pharynx [5,26,27].

The determination of the course of infection often necessitates several samplings, especially at the beginning of the infection, when a battle between the cat’s immune system and the virus can prevail; but what the clinician usually sees is only a snapshot (one-time consultation) [2]. Determining the course of infection is therefore based on an assumption. For example, regressively infected animals can become progressively infected because of reactivation of the virus [1]. In the present study, there was one cat that was negative in the p27 antigen ELISA but tested positive for viral RNA in the blood. The reason why this cat was assigned to a progressive course was that all other laboratory results indicated a progressive course of FeLV infection (viral-RNA-positive in blood, proviral-DNA-positive, anti-p15E antibody-positive). It is possible that the p27 antigen ELISA in this cat was false-negative, although this is unlikely due to several repetitions of the ELISA. Another explanation could be that the cat was at the beginning of the FeLV infection and the p27 antigen could not yet be detected. The cat had a high proviral load in blood (8503 copies/PCR reaction). However, it should be noted that, during the early stages of infection, both progressively and regressively infected cats can exhibit a high proviral load [4]. In the present study, as well as in a recently published study, it was shown that progressively infected cats had higher anti-p15E antibody levels compared to regressively infected cats [28]. The cat classified as progressively infected had a high antibody level of 92%, which is also indicative of a progressive course.

Studies have shown that, once the course of infection is established, progressively infected cats, in contrast to regressively infected cats, have a higher proviral load, and the proviral load in progressively infected cats remains consistently high after an initial phase [29]. In the present study, progressively infected cats had a significantly higher proviral load compared to regressively infected cats. This suggests that the determination of proviral DNA in blood by PCR alone can provide an indication of the course of the infection. A study from Switzerland showed that cats that tested negative for p27 antigen but positive for proviral DNA in blood had significantly lower proviral load (by a factor of 300) [29]. However, it was impossible to draw a cut-off that distinguished all regressively infected cats from progressively infected cats; there was some overlap the proviral DNA loads between the two groups of cats. Thus, it is important to repeat the test at a later timepoint and to initiate further diagnostic measures (p27 antigen, viral RNA), especially in case of unclear results, to allow a clear assignment to a course of infection. Especially in naturally infected cats, the assignment of a course is not always possible, as the time of infection is usually unknown and the amounts of proviral DNA do not differ between progressively and regressively infected cats during early infection [4].

The reason for the different distribution of FeLV course in the individual countries in the present study is still unclear. In general, several subgroups of exogenous FeLV (exoFeLV) (FeLV-A, -B, -C, -D, -E, and -T) are distinguished in addition to endogenous FeLV (enFeLV). Multiple studies provided evidence of an association between enFeLV loads and the replication of FeLV-A. In one study, it was observed that cats with high enFeLV loads were more likely to develop progressive FeLV infection, while cats with low enFeLV loads had a lower risk [8,23,24,30]. There might be differences between the FeLV subtypes in the respective countries. A study from the USA in 2018 showed that cats with a progressive course and a higher proviral load were significantly more likely to be FeLV-B-positive. In contrast, regressive, abortive, and uninfected cats had higher levels of enFeLV [31]. The properties of the cat, such as age or genetic background, including the endogenous viral load, might also play a role in the development of the respective courses in the present study. In order to strengthen this assumption, further investigations, such as genetic analyses, would be necessary.

There are only a few studies that determined FeLV prevalence considering all courses of FeLV infection. In a study performed in Southern Germany in 2012, 7.5% (37/495) of cats were infected with FeLV. Out of these, 1.8% (9/495) of cats were progressively infected (p27 antigen-positive), 1.2% (6/495) were regressively infected (p27 antigen-negative, provirus-positive), and 9.2% (22/246) of unvaccinated cats were abortively infected with FeLV (anti-p45 antibody-positive; >25%). The presence of focally infected cats was not investigated [12]. The prevalence of progressively and regressively infected cats was comparable to the results from Germany in the present study; however, the proportion of abortively infected cats in the present study was 3.5%, in contrast to 9.2% previously. One reason for this difference could be that the criteria for the classification of an abortive course were chosen differently. In the German study, all SU antibody-positive cats were assigned to an abortive course of infection. In the present study, cats had to be anti-SU antibody-positive and anti-p15E antibody-positive to be considered abortively infected. Cats with anti-SU antibody concentrations >25% of the positive control were defined as positive [12]. This cut-off was chosen because SPF cats can have antibody concentrations of up to 25% because of unspecific reactions or antibodies against enFeLV (H Lutz, E Boenzli, data not published, personal communication). Anti-p15E antibody levels >16.3% (relative optical density (ROD) value 0.163) of the positive control were therefore considered positive, as defined previously for field cats when establishing the p15E ELISA [10]. A recently published study showed that, in Australia, 57.4%, and in Germany, 8.2% of FeLV-unexposed cats had anti-p15E antibody tires >16.3%. These findings highlight that the cut-off value of the p15E laboratory ELISA should be considered critically and should be reconsidered [28]. Anti-SU antibodies can be strongly increased after vaccination against FeLV, depending on the vaccine used, as well as after FeLV infection. Cats vaccinated against FeLV using canarypox vectored vaccines might not develop detectable antibody levels before coming into contact with FeLV [18,32]. In a study from Germany, anti-SU antibodies could not be detected in 64.3% of cats vaccinated against FeLV [12]. In contrast, cats vaccinated with a recombinant FeLV subunit vaccine containing p45 (unglycosylated SU) showed a high anti-SU antibody response [4,33]. Curiously, there were more vaccines used in Italy than in Portugal, Germany, and France, yet the FeLV prevalence was higher in Italy. It should, however, be noted that the FeLV vaccination status of many cats was unknown and therefore the number of cats vaccinated against FeLV was likely much higher in all countries. Since the vaccination status of many cats was unknown in the present study, anti-p15E antibodies were additionally selected to assign the cats to an abortive course of infection. In a study carried out in Switzerland, serum samples from 294 cats were used to test the suitability of the detection of anti-p15E antibodies for the diagnosis of FeLV infection [10]. Anti-p15E antibodies were found to be useful to differentiate infected from uninfected animals, suggesting that anti-p15E antibodies indicate previous infection rather than vaccination [10,20]. In the present study, cats with a progressive course had higher anti-p15E antibody levels compared to cats with regressive, abortive, or focal infection. Furthermore, a positive correlation between the concentration of p27 antigen in blood and the level of anti-p15E antibody levels in the blood was found. Regressively infected cats had high anti-whole virus antibody levels, while progressively infected cats had low concentrations of anti-whole virus and anti-SU antibodies, except for a few outliers. This suggests that the presence of anti-whole virus and anti-SU antibodies is more likely to be a sign of protection against FeLV infection when compared to anti-p15E antibodies. It is still unclear whether the presence of anti-p15E antibodies also corresponds to protection against FeLV and how long these antibodies are detectable (after vaccination and infection). In the present study, the presence of anti-p15E antibodies was more likely a marker for viral replication. It is also possible that cats that were classified as progressively infected and that had high anti-whole virus antibodies could be in a phase of switching to regressive infection if the antibodies were constantly high [34]. Since all cats were only sampled on a single date, no statement can be made about the long-term course.

Therefore, the most important limitation of the present study was that the FeLV status was determined only based on one blood sample. The second limitation was the presence of selection bias, as only cats in which blood was taken any way for diagnostic reasons were included. This unavoidable preselection might have influenced FeLV prevalence. However, it is important to note that these data provide insights into the FeLV situation among cats admitted to veterinary practices in the respective countries.

## 5. Conclusions

In conclusion, this is the first study that determined the prevalence of FeLV in four European countries considering all courses of FeLV infection (progressive, regressive, abortive, focal (atypical)). When all courses of infection were taken into account, the overall FeLV prevalence was still very high, especially in Italy and Portugal (Southern Europe). The number of infected cats in Germany and France (Western Europe) was also still remarkably high. Therefore, the FeLV status of each cat should be known. If a cat cannot be assigned to a course of infection, follow-up testing and the use of further laboratory diagnostic methods can be helpful. Even though FeLV prevalence in the tested countries decreased in previous years, further steps (testing, separation of infected cats, and vaccination) are needed to control infection risk and reduce prevalence further.

## Figures and Tables

**Figure 1 viruses-15-01718-f001:**
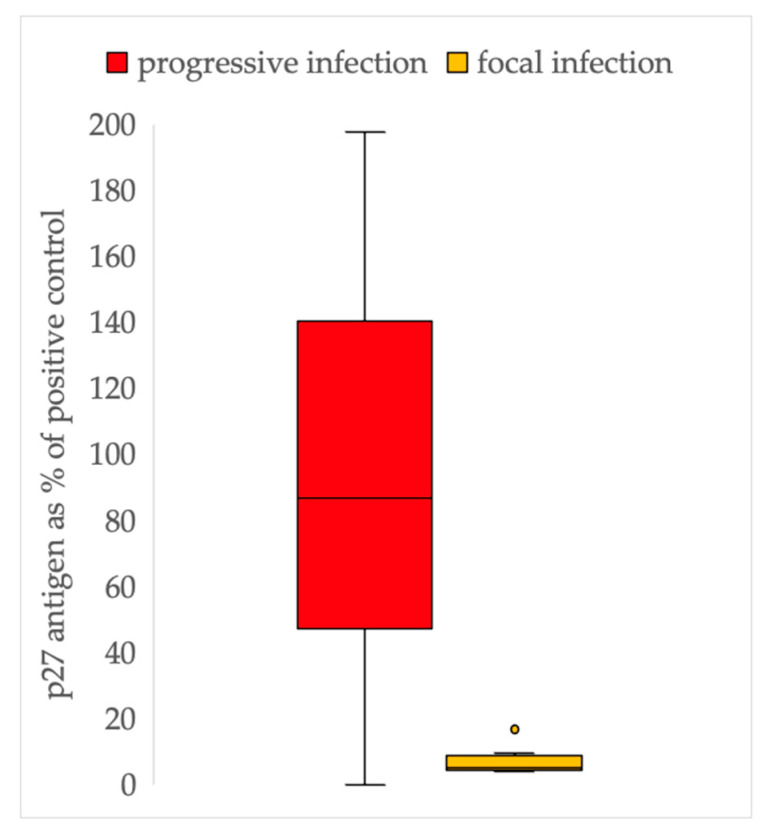
Results of p27 antigen enzyme-linked immunosorbent assays (ELISA) in cats with progressive (p27 antigen-positive, provirus-positive) and focal feline leukaemia virus infections (p27 antigen weakly positive, provirus-negative). Progressively infected cats had significantly higher antigen concentrations (median: 91%, range: 5–197%, *p* < 0.001) than cats with focal infection (median: 6%, range: 4–20%).

**Figure 2 viruses-15-01718-f002:**
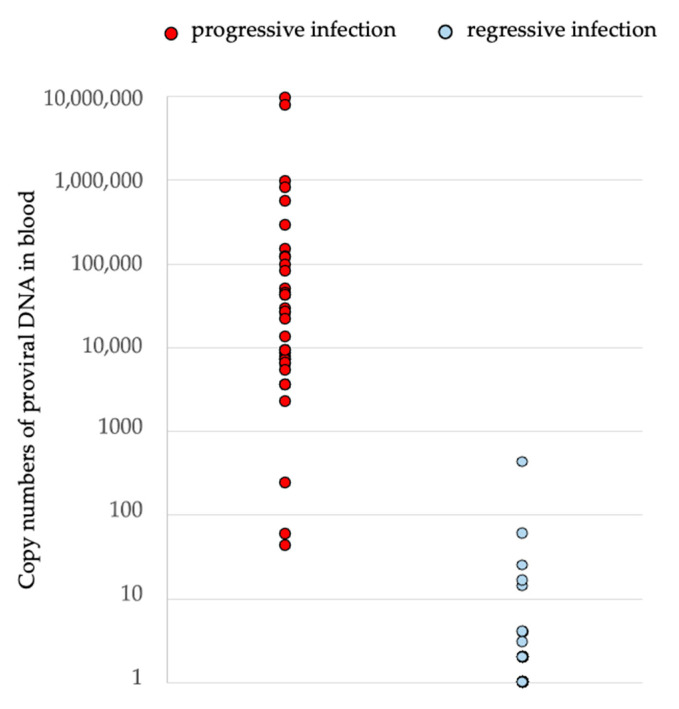
Proviral load in progressively (p27 antigen-positive, viral RNA in saliva and/or blood detectable, provirus-positive) and regressively infected cats (p27 antigen-negative, provirus-positive). Cats with progressive infection had significantly higher copy numbers (median: 21,543, range: 43–9,501,447) compared to cats with regressive infection (median: 1, range: 1–428) (*p* < 0.001).

**Figure 3 viruses-15-01718-f003:**
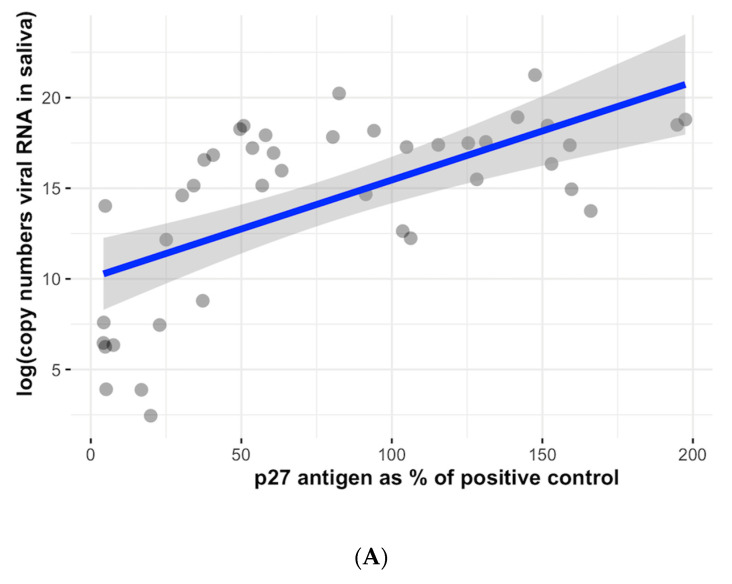
Correlation between p27 antigen concentration in blood and viral RNA loads in blood and saliva. The grey shading indicates 95% confidence intervals. (**A**) A strong significant correlation was observed between the p27 antigen concentration and viral RNA load in saliva (Spearman correlation = 0.65; *p* < 0.001). (**B**) A significant correlation was also observed between the p27 antigen and viral RNA load in the blood (Spearman correlation = 0.38; *p* = 0.027). RNA, ribonucleic acid.

**Figure 4 viruses-15-01718-f004:**
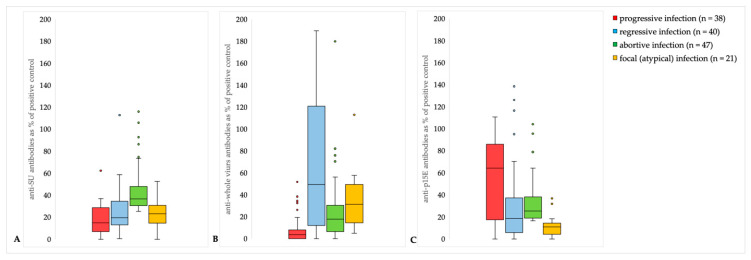
Box plots of anti-whole virus (FL-74), anti-SU (p45), and anti-p15E antibodies in progressively (red), regressively (blue), abortively (green), and focally (yellow) infected cats. (**A**) Cats with abortive infection had significantly higher anti-SU (p45) antibody levels (median: 37%, range: 25–116%) compared to cats with progressive (median: 15%, range: 0–62%) (*p* < 0.001), regressive (median: 20%, range: 1–113%) (*p* < 0.001), and focal (median: 23%, range: 6–53%) (*p* < 0.001) infection. (**B**) Regressively infected cats had significantly higher anti-whole virus antibody levels (median: 49%, range: 0–190%) compared to progressively (median: 4%, range: 0–63%) (*p* < 0.001) and abortively (median: 18%, range: 0–82%) (*p* = 0.011) infected cats. Progressively infected cats had significantly lower antibody tires compared to abortively infected (*p* = 0.005) and focally infected cats (median: 23%, range: 5–123%) (*p* ≤ 0.001). (**C**) Progressively infected cats had significantly higher anti-p15E antibody levels (median: 64%, range: 0–121%) compared to abortively (median: 25%, range: 17–114%) (*p* < 0.001) infected cats. Abortively infected cats had significantly higher antibody levels compared to regressively (median: 19%, range: 0–152%) (*p* < 0.001) and focally infected cats (median: 11%, range: 0–40%) (*p* = 0.002).

**Figure 5 viruses-15-01718-f005:**
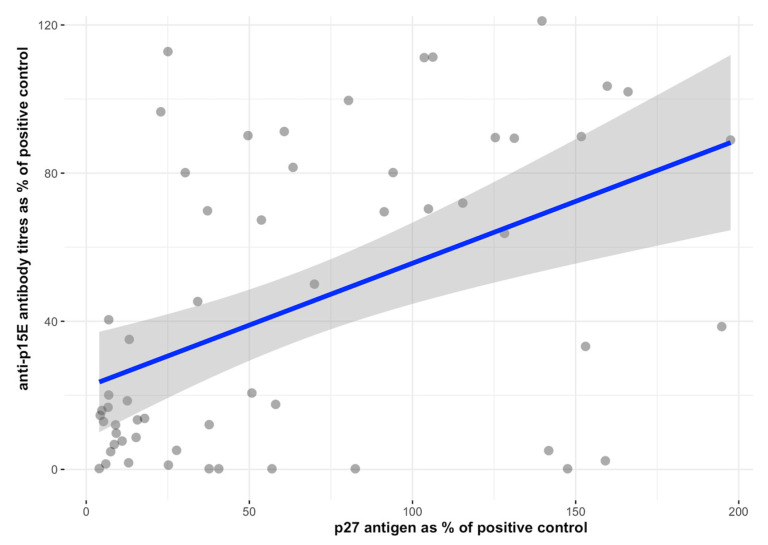
Correlation between anti-p15E antibody levels in blood and p27 antigen levels in blood. A significant positive correlation was observed between the anti-p15E antibody concentration and p27 antigen concentration in blood (Spearman correlation = 0.431; *p* ≤ 0.001.

**Table 1 viruses-15-01718-t001:** Courses of feline leukaemia virus infection (FeLV) and the expected test results. Adapted from the European Advisory Board on Cat Diseases FeLV diagnostic tool [15].

	Progressive	Regressive	Abortive	Focal (Atypical)
	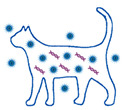	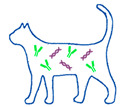	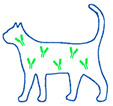	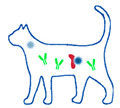
p27 AG	positive	negative	negative	(weakly) positive
viral RNA	positive	negative	negative	negative or positive
proviral DNA	positive	positive	negative	negative
anti-SU/whole virus AB	negative	(usually) positive (variable titre)	positive	(usually) positive (variable titre)
anti-p15E AB	negative or positive	(usually) positive (variable titre)	positive	(usually) positive (variable titre)

AG, antigen; RNA, ribonucleic acid; DNA, deoxyribonucleic acid; AB, antibody.

**Table 2 viruses-15-01718-t002:** Signalment and history of the cats included in the study.

Variables	Modalities	Italy n = 269 (%)	Portugal n = 240 (%)	Germany n = 318 (%)	France n = 107 (%)	Total n = 934 (%)
age	kitten (<1 year)	80 (30)	47 (20)	80 (25)	6 (6)	213 (23)
	young adult (1–6 years)	98 (36)	55 (23)	122 (38)	22 (20)	297 (32)
	mature adult (7–10 years)	51 (19)	20 (8)	56 (18)	13 (12)	140 (15)
	senior (>10 years)	33 (12)	8 (3)	54 (17)	20 (19)	115 (12)
	unknown	7 (3)	110 (46)	6 (2)	46 (43)	169 (18)
sex	female	126 (47)	88 (37)	154 (48)	35 (33)	403 (43)
	male	139 (52)	68 (28)	161 (51)	24 (22)	392 (42)
	unknown	4 (1)	84 (35)	3 (1)	48 (45)	139 (15)
neutering status	intact	77 (29)	13 (5)	60 (19)	5 (4)	155 (16)
	neutered	176 (5)	131 (55)	218 (69)	51 (48)	576 (62)
	unknown	16 (6)	96 (40)	40 (12)	51 (48)	203 (22)
breed	mixed breed cat *	236 (88)	133 (55)	195 (61)	48 (45)	612 (66)
	purebred cat	28 (10)	17 (7)	121 (38)	3 (3)	169 (18)
	unknown	5 (2)	90 (38)	2 (1)	56 (52)	153 (16)
housing status	outdoor only	65 (24)	50 (21)	61 (19)	2 (2)	178 (19)
	in- and outdoor	55 (21)	27 (11)	26 (8)	1 (1)	109 (12)
	indoor only	138 (51)	70 (29)	141 (44)	2 (2)	351 (37)
	shelter	0 (0)	0 (0)	74 (23)	0 (0)	74 (8)
	unknown	11 (4)	93 (39)	16 (6)	102 (95)	222 (24)
cats in household	1 cat	79 (29)	11 (5)	131 (41)	1 (1)	222 (24)
	2–4 cats	90 (33)	26 (11)	68 (22)	0 (0)	184 (20)
	≥5 cats	42 (16)	41 (17)	33 (10)	0 (0)	116 (12)
	shelter, colony, stray	26 (10)	8 (3)	76 (24)	0 (0)	110 (12)
	unknown	32 (12)	154 (64)	10 (3)	106 (99)	302 (32)
history of illness	healthy	170 (63)	112 (47)	95 (30)	0 (0)	375 (40)
	pre-existing illness/sick	88 (33)	27 (11)	177 (56)	1 (1)	293 (32)
	unknown	11 (4)	101 (42)	46 (14)	106 (99)	264 (28)
FeLV vaccination status	vaccinated	34 (13)	11 (5)	24 (7)	12 (11)	81 (9)
	not vaccinated	224 (83)	115 (48)	231 (73)	3 (3)	573 (61)
	unknown	11 (4)	114 (47)	63 (20)	92 (86)	280 (30)

* European Shorthair, European Longhair, Domestic Shorthair, Domestic Longhair; n, number of cats.

**Table 3 viruses-15-01718-t003:** Number of cats in which the feline leukaemia virus (FeLV) vaccination status was known that had received different FeLV vaccines in the individual countries.

Vaccine *	Italy n = 269 (%)	Portugal n = 240 (%)	Germany n = 318 (%)	France n = 107 (%)	Total n = 934 (%)
Purevax^®^ FeLV	24 (9)	4 (2)	15 (5)	-	43 (5)
Leucogen^®^	2 (1)	1 (0)	4 (1)	-	7 (0)
Versifel^®^ FeLV	2 (1)	1 (0)	-	-	3 (0)
Fevaxyn^®^ Pentofel	-	-	-	1 (1)	1 (0)
unknown vaccine	6 (2)	5 (2)	5 (1)	11 (10)	27 (3)
total	34 (13)	11 (4)	24 (7)	12 (11)	81 (9)

* In total, four different FeLV vaccines were used. The vaccines used against FeLV included recombinant canarypox virus (vCP97) vaccine (Purevax^®^ FeLV, Boehringer Ingelheim Vetmedica GmbH, Rohrdorf, Germany), a monovalent FeLV subunit vaccine (Leucogen^®^, Virbac Animal Health, Carros, France), an inactivated whole virus vaccine containing FeLV-A, FeLV-B, and FeLV-C (Versifel^®^ FeLV, Zoetis Animal Health, Parsippany-Troy Hills, NJ, USA), and an inactivated feline leukaemia virus vaccine (Fevaxyn^®^ Pentofel, Zoetis, Ottignies-Louvain-la-Neuve, Belgium).

**Table 4 viruses-15-01718-t004:** Number and percentage of positive test results of all cats included in the study from Italy, Portugal, Germany, and France.

Test	Italy n = 269 (%)	Portugal n = 240 (%)	Germany n = 318 (%)	France n = 107 (%)	Total n = 934 (%)
p27 AG	27 (10.0)	13 (5.4)	15 (4.7)	3 (2.8)	58 (6.2)
viral RNA	21 (7.8)	9 (3.7)	6 (1.9)	2 (1.9)	38 (4.1)
proviral DNA	33 (12.3)	29 (12.1)	10 (3.1)	6 (5.6)	78 (8.3)
anti-SU AB	53 (19.7)	74 (30.8)	66 (20.7)	15 (14.0)	208 (22.2)
anti-whole virus AB	33 (12.3)	35 (14.6)	42 (13.2)	23 (21.5)	133 (14.2)
anti-p15E AB	57 (21.2)	54 (22.5)	45 (14.1)	39 (36.4)	195 (19.8)

AG, antigen; RNA, ribonucleic acid; DNA, deoxyribonucleic acid; AB, antibodies; SU, surface unit; n, number of cats.

**Table 5 viruses-15-01718-t005:** Prevalence of the different courses of feline leukaemia virus (FeLV) infection in Italy, Portugal, France, and Germany.

Course of Infection	Italy n = 269 (%)	Portugal n = 240 (%)	Germany n = 318 (%)	France n = 107 (%)	Total n = 934 (%)
progressive	21 (7.8)	9 (3.8)	6 (1.9)	2 (1.9)	38 (4.1)
regressive	12 (4.5)	20 (8.3)	4 (1.3)	4 (3.7)	40 (4.3)
abortive	17 (6.3)	16 (6.7)	11 (3.5)	3 (2.8)	47 (5.0)
focal (atypical)	7 (2.6)	4 (1.7)	9 (2.8)	1 (0.9)	21 (2.3)
total infected	57 (21.2)	49 (20.4)	30 (9.4)	10 (9.4)	146 (15.6)
unclassified ^1^	6 (2.2)	2 (0.8)	4 (1.3)	8 (7.5)	20 (2.1)

^1^ Samples that were p27 antigen-negative, provirus-negative, anti-p15E antibody-positive, but anti-SU antibody-negative were considered “unclassified”.

## Data Availability

The authors confirm that the datasets analysed during the study are available from the corresponding author upon reasonable request.

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
