# Peer review of "Prevalence of Different Courses of Feline Leukaemia Virus Infection in Four European Countries"

_viruses, 2023, doi:10.3390/v15081718_

Round 1
Reviewer 1 Report
This paper describes the prevalence of different forms of feline leukemia virus (FeLV) status of cats presented to various types of veterinary clinics in four European countries. It combines a descriptive study of serologic testing with an attempt at epidemiologic analysis, which falls short. It is recommended that assistance be obtained from a biostatistician/veterinary epidemiologist because the analysis methods and presentation of results were incompletely described and not appropriate. Presumably there was not one “univariate and multivariate logistic analysis performed” (line214). Quite a few parameters were examined; there was no comment about correction for multiple comparisons (such as Bonferroni). Be aware that the p-value for significance should never include 0.05, as on line 217 – that is a red-flag for reviewers!
The comparisons between forms of FeLV were described in a confusing fashion. The purpose was not clear. For example, in lines 257-8, it seems the reference category was not unaffected cats but focally infected cats. Is this correct? Seems a bit unusual. Comparing every category to every other category is mathematically tempting but not meaningful. Again, please consult an epidemiologist. In that sentence, move the p-value to the end of the sentence in which comparisons were made “with” and not “to” the reference category; there is no need to specify the type of p-value if the description was made clear in the Methods section. More difficulties requiring statistical assistance are described below. Using software and being able to correctly state the basis for testing and the results are different things; again, obtain assistance with all this.
In Tables 2, 4, and 5, combine the columns for variables, such as sex, with options listed below. The “total” columns in the tables are not really useful, as no analysis was done with these values. In Tables 4 and 5, add columns for the numbers of cases and controls that went into the logistic regression analyses. Which were the controls for the analysis of each form? Since there were so many unknowns, it is possible that all the significant results are based on very few cats.
Such results are not generalizable to the wider world of cats and must be reported with the caveat that they were found *in this study* in the Discussion and particularly in the Conclusions. It may be appropriate to remove the logistic regression analyses from this manuscript, since so few cats appear in each of the various categories being analyzed and these analyses appear to have been an afterthought.
Other comments:
Recommend changing “pedigree” to “purebred” – in the US, pedigree is most commonly used for dogs with known breeding history (papers) identifying predecessors, thus pedigree. This information is generally not known for cats, especially for cats in shelters. Rather, “purebred” is often used for visually identifiable breeds, such as Siamese or Himalayan, and otherwise, cats are mixed breeds, such as Domestic Shorthair, Domestic Medium-hair, and Domestic Longhair.
Recommend care in use of singular and pleural nouns and verbs. For example, in lines 368-9, “results … was” is not correct. See more below.
Specific comments:
Line 26: Consider re-ordering Portugal and Italy to be consistent with the rest of the presentation.
Line 52: Recommend changing “a FeLV cat” to “an FeLV cat.”
Line 53: Recommend changing “e.g.,” to “such as” to be consistent with the rest of the sentence.
Line 66: Recommend “briefly” instead of “shortly” and “recurs” instead of “reoccurs” in this sentence.
Lines 73 and 84: The word “prevalence” should always be singular.
Line 96-7: Recommend changing this sentence to “Cats from which blood was collected for various reasons were included.”
Line 104: No comma is required after “Both.”
Line 209: Recommend adding “The” before “Mann-Whitney U” at the start of this sentence. To be correct, this test was used to compare each of the FeLV forms with the unaffected cats, yes? Recommend saying so explicitly, otherwise this reads as though all were compared at once to each other, and the Results did not make this clear at all.
Line 236 and Table 3: The first column is “Test” and the title should be “Results of testing performed…”
Lines 240, 241, Table 4, and throughout: Recommend changing “course” to “form” or similar, since as noted, cats may move between forms of infection. “Course” implies moving in one direction, so does not apply here. Use the terminology of cited literature to be consistent.
Lines 257, 264, and throughout: Consider reducing the number of the significant figures presented in p-values; four seems a lot.
Lines 298, 299, and beyond: Now results were presented as “median” and “average” – why? Non-normally distributed parameters must be described as median (range) rather than mean (± standard deviation) – and “mean” and “average” are the same. Please consult with the epidemiologist about this.
Line 304: Presumably, “tire” is meant to be “titre” here.
The heading and introductory sentences in Section 3.2 should appear in Methods, not Results. However, since the reference category used for these analyses was not specified in either section, it is unclear how the reader is to interpret the results.
In the Discussion section, the first sentence in the first paragraph requires a period. That first paragraph repeats the Introduction – condense and reorganize to highlight the results of this study in comparison with other studies, rather than simply repeating other literature. The second sentence in the first paragraph should appear in the Methods and not repeated here. Fix the sentence “The most important results of the study was still” since this implies more than one important result – check that nouns and verbs match – one item is most important. The following several paragraphs of the Discussion appear to discuss the forms of FeLV in great detail – however, this manuscript is not a review of these forms and so this information should be provided only to place the results of this study into the context of the literature. However, the statement in lines 385-6 that “FeLV prevalence is known to be higher in animal shelters” citing a study of two shelters in the UK seems a substantial stretch, since that study only described prevalence in shelters – there was no outside comparison group. Thus, higher compared with what?
Line 504: Recommend correcting the grammar from “could be identified” to “were” and adding “in this study,” in case this reviewer believes the analysis conducted.
Line 522: Presumably, “bias” was intended, as in “selection bias.” Please, discuss terminology with the epidemiologist to be consulted.
Conclusions:
As noted, conclusions should not generalize results to the wider world. In a well-conducted study, the conclusions can only state that certain results were found in this population. A suggestion that clinicians consider these results in their practices can be made, but that is as far as authors can go. Use past verb tense throughout, because by the time a reader sees this paper, the contents refer to the past study.
Line 528: Recommend “When” rather than “If” at the start of the sentence, if the reader believes the analysis
Line 530: Delete “also” and add “other” before “Western European” in this sentence.
Line 534: Recommend changing “could be reduced in previous years” to “were” if that is what was meant – this was all very confusing.
Comments regarding language have been included above.
Reviewer 2 Report
This is an excellent example of a cross country study investigating the prevalence of feline leukaemia virus in Portugal, German,. Italy and France. The study uses a variety of good serological and molecular methods to diagnose infection, and categorises them as per the cat advisory board standards (although this is possibly slightly messy from the board).
The manuscript is well written, clear and contains a large number of animals, with some interesting results and suggestions within it
As such, I only have a few very minor comments which are detailed below.
Line 66- only lasts shortly sounds a bit awkward- perhaps is only transient? Or is short lived?
Line 90-92- you could consider removing this as its repeats the aims, but that is up to the authors
Table 2- is there a difference within the vaccine used in the different countries? If so that may be worth stating with a bit of detail
Section 3.5- I found this a bit heavy going. Would it work better as a table perhaps? Again up to the authors and just a suggestion
Line 367- I think some punctuation maybe needed between (atypical)) and cats?
Line 378- would course of infection sound better than in infection?
Line 507- perhaps in a recent pan- European rather than recently? Maybe better reworded?
Round 2
Reviewer 1 Report
The revised manuscript is much improved. Thank you for obtaining epidemiologic assistance and taking the suggestions to heart. It is always difficult to remove material, so your response is appreciated.
A few comments on the revised manuscript:
Lines 214-216: This sentence requires a comma or other reorganization. Perhaps this is meant to be: “Due to non-normally distributed data, a non-parametric test was used…”
The information in Table 3 is interesting but not discussed. Recommend putting all but the first sentence (title) into a “Note” below the table. Curiously, there were more vaccines used in Italy and Portugal than in Germany and France, yet the FeLV prevalence was higher in those countries (though overall, few vaccines were used in cats in this study). Recommend adding some commentary about this, and perhaps condensing the information from the table into text form.
Line 426: Suggest changing “a high number” to “greater number” or similar, since “high number” suggests the reader knows what a “usual number” may be. Really, this is meant to say, a larger number than expected, yes?
Line 512: Suggest changing “might” to “may” in this sentence.
Lines 542-4: Is the information in this sentence *always* correct? Suggest changing “do not develop antibody levels” to “may not develop antibody levels” if appropriate.
Line 606: Suggest changing “was reduced” to “decreased” in this sentence.
Comments appear above.
